# Vision-Based Trajectory Reconstruction in Human Activities: Methodology and Application

**DOI:** 10.3390/s25247577

**Published:** 2025-12-13

**Authors:** Jasper Lottefier, Peter Van den Broeck, Katrien Van Nimmen

**Affiliations:** Structural Mechanics, Department of Civil Engineering, KU Leuven, B-9000 Ghent, Belgium; peter.vandenbroeck@kuleuven.be (P.V.d.B.); katrien.vannimmen@kuleuven.be (K.V.N.)

**Keywords:** empirical trajectory reconstruction, drone-based object tracking, crowd dynamics measurement frameworks, integrated human movement measurements

## Abstract

Modern civil engineering structures, such as footbridges, are increasingly susceptible to vibrations induced by human activities, emphasizing the importance of accurately assessing crowd-induced loading. Developing realistic load models requires detailed insight into the underlying crowd dynamics, which in turn depend on the coordination between individuals and the spatial organization of the group. A deeper understanding of these human–human interactions is therefore essential for capturing the collective behaviour that governs crowd-induced vibrations. This paper presents a vision-based trajectory reconstruction methodology that captures individual movement trajectories in both small groups and large-scale running events. The approach integrates colour-based image segmentation for instrumented participants, deep learning–based object detection for uninstrumented crowds, and a homography-based projection method to map image coordinates to world space. The methodology is applied to empirical data from two urban running events and controlled experiments, including both stationary and dynamic camera perspectives. Results show that the framework reliably reconstructs individual trajectories under varied field conditions, applicable to both walking and running activities. The approach enables scalable monitoring of human activities and provides high-resolution spatio-temporal data for studying human–human interactions and modelling crowd dynamics. In this way, the findings highlight the potential of vision-based methods as practical, non-intrusive tools for analysing human-induced loading in both research and applied engineering contexts.

## 1. Introduction

### 1.1. Problem Statement

Modern civil engineering structures are increasingly designed to be light and slender. While this trend offers structural and economic advantages, such designs also increase the susceptibility to low-frequency dynamic excitations. In the case of footbridges, walking and running loads have been shown to induce vibrations that can exceed acceptable comfort levels [1,2]. These issues are particularly pronounced under resonance conditions, where the degree of synchronization between pedestrians and the structure becomes the key factor governing the dynamic response.

Whereas traditional static safety checks, commonly applied in the design of conventional civil engineering structures, focus on the ultimate limit state, vibration serviceability, which concerns user comfort, often governs the design of footbridges [3]. Accurate assessment of vibration comfort, typically expressed in terms of the structural accelerations [1,4], requires a detailed characterization of the expected dynamic loading. Human-induced forces must therefore be validated on vibrating structures, such as lively footbridges, to capture Human–Structure Interaction (HSI) effects [5,6]. Furthermore, when extending individual force models to group loading, Human–Human Interaction (HHI) must also be considered [7].

Studying HHI is methodologically challenging, as it requires detailed spatio-temporal gait data and the evolving positions of all participants. These difficulties contribute to the scarcity of empirical in-field studies. Due to the lack of comprehensive experimental data, current vibration design provisions often assume synchronized pedestrian behaviour in dense crowds, although supporting evidence remains limited [8,9,10,11,12,13]. The topic is even less studied for running, with empirical data on crowd densities and the thresholds at which running becomes impractical largely absent. Furthermore, although some interaction is expected in congested environments, such as during mass-participation running events, it is unclear whether this necessarily leads to synchronized movement. Consequently, no established framework exists to evaluate vibration serviceability for running.

To address the lack of empirical data, in-field observations of collective running behaviour are required. While spatio-temporal gait parameters can be measured using existing techniques, such as motion capture systems, capturing the time-varying orientation of multiple individuals in outdoor environments remains challenging. This study therefore develops and applies a measurement framework capable of recording such information under realistic conditions. Although the primary focus is on running, the framework is sufficiently versatile to accommodate other forms of pedestrian activity, including walking.

### 1.2. Experimental Data Collection to Study Crowd Dynamics

Understanding crowd dynamics requires accurate measurements of the collective movement, for example by reconstructing the trajectories of individual participants. Person-tracking sensors are generally classified into two categories: (1) wearable and (2) optical sensors. For small groups, individual tracking is typically facilitated by equipping each runner with a wearable motion sensor or visual markers. In contrast, for large, high-density crowds, such instrumentation becomes impractical, necessitating alternative approaches, predominantly based on optical sensors.

Wearable sensors, such as IMUs attached to runners, can be combined with dead reckoning, where accelerometer and gyroscope data are integrated to estimate position and orientation over time [14]. Although conceptually simple, this approach is highly sensitive to noise and drift, leading to cumulative errors that grow quadratically with time. Drift is typically corrected using absolute positioning systems such as GPS, yet the spatial resolution of GPS, usually on the order of meters, remains insufficient for detailed motion tracking. Comparable limitations exist for WiFi-based localization methods, which are further constrained in outdoor environments [15,16].

Optical sensors, such as RGB cameras, provide an alternative by enabling vision-based motion tracking, either marker-based or markerless. Indirect force estimation using optical motion capture systems with light-emitting or retro-reflective markers is a well-established approach in controlled environments. Markerless, computer vision–based methods extend this principle by tracking general image features across video frames, allowing applications in more natural or outdoor settings. In civil engineering, vision-based techniques have demonstrated performance comparable to conventional accelerometer-based approaches in structural health monitoring and modal analysis [17,18]. Regarding pedestrian tracking, early attempts relied on (semi-)manual tracking of visible features across video frames [19,20,21], such as the midpoint between the eyes [22,23]. With advances in computer vision and person detection, feature extraction became more systematic, and the introduction of deep learning largely removed the need for handcrafted inputs. State-of-the-art models (semi-)automatically detect, segment and track individuals in video streams using textual prompts [24]. Advances in fields such as autonomous driving have accelerated these developments, establishing vision-based techniques as a powerful tool for analysing large and dense pedestrian flows.

Several vision-based frameworks have been developed for motion tracking and human pose estimation. *PeTrack* [25] is a semi-automated system that reconstructs trajectories from overhead footage using background subtraction, feature matching and camera calibration with reference markers [26]. Open-source tools such as *OpenPose* [27] and *DeepLabCut* [28] provide markerless full-body pose estimation based on Convolutional Neural Networks (CNNs) [29], while *AlphaPose* [30] extends this capability to multi-person whole-body pose tracking.

Although existing software packages offer robust solutions for motion tracking, their performance is often constrained by specific acquisition conditions, such as controlled laboratory settings, (semi-)static camera positions, high-resolution imagery, or footage captured primarily from side or oblique angles. As these conditions are neither preferred nor guaranteed in this work, most available tools are not directly applicable. To address this gap, a streamlined framework is developed that builds on recent advances in multi-object detection and tracking, adopting a markerless approach tailored to overhead imagery. Furthermore, because optical sensors are used, established photogrammetric methods are applied to map image coordinates into world coordinates.

### 1.3. Contribution of the Present Work

This study presents a vision-based trajectory reconstruction methodology, able to capture human walking and running activities from a single camera perspective. Unlike existing approaches that rely on laboratory conditions, controlled instrumentation, multi-camera setups, or oblique camera perspectives, the proposed framework is tailored to realistic environments using overhead RGB video footage. The method is directly applied to empirical running data, thereby demonstrating its applicability. The reconstruction methodology inputs raw video frames and outputs the trajectories, using multiple steps such as detection and tracking. The motivation for the presented paper is to establish a complete workflow for practical applications, given a level of accuracy equal to typical human body radii (0.1 to 0.15 m).

The methodology accommodates both dynamic and stationary camera perspectives. A moving camera perspective enables detailed in-field measurements of small groups in open environments, while a stationary perspective supports large-scale crowd monitoring. For smaller groups, where participant instrumentation is feasible, a colour-based image segmentation strategy is applied. This processing pipeline builds upon the work of Van Hauwermeiren et al. [31], originally developed for fixed multi-camera networks, and is substantially extended by adapting it to a single dynamic camera. Key modifications include the reformulation of the detection and tracking pipeline for dynamic viewpoints and the integration of photogrammetric principles for projection to world space. For large crowds, where individual instrumentation is impractical, a complementary methodology is proposed based on deep learning–driven object detection, ensuring scalability across a wide range of experimental conditions.

Together, these contributions provide a versatile, non-intrusive framework that is able to monitor both small groups as well as large-scale running events. Such datasets are vital in the analysis of human–human interactions.

Importantly, this study addresses a gap in the current literature concerning the availability of a practical, end-to-end workflow for extracting human trajectories under highly dense in-field measurement conditions. The objective is not to advance state-of-the-art object detection or tracking methods, but to integrate established approaches into a coherent and functional pipeline. The contribution lies in demonstrating how raw video data, acquired from a single optical sensor from nadir perspectives, either stationary or moving, can be processed to reconstruct individual trajectories in both controlled and uncontrolled measurement scenarios that reflect real-world conditions.

The remainder of this paper is organized as follows. Section 2 introduces the experimental program, complemented by a brief overview of relevant applied computer vision and photogrammetry techniques. The presented measurements form the basis for discussing and validating the trajectory reconstruction methodology in Section 3. Section 4 presents the results together with a detailed discussion of systematic reconstruction errors while Section 5 provides the concluding remarks.

## 2. Materials and Methods

The measurement framework is designed to capture individual walking or running trajectories from a single camera perspective. An experimental campaign was conducted to record various running scenarios to which the proposed generic vision-based trajectory reconstruction approach is applied. Spatio-temporal trajectories are reconstructed for each person within the Field of View (FoV) of the camera. Accordingly, an aerial nadir perspective is proposed to reduce occlusions and mitigate reconstruction errors.

### 2.1. Experimental Program

The experimental program comprises two distinct measurement conditions, as illustrated in Figure 1. In the first scenario (Figure 1a), a fixed-size group of runners is tracked while moving in an open space (athletics track). As a single camera is used, its position changes parallel to the group, enabling the reconstruction of trajectories for the same individuals throughout the entire running event. In the second scenario (Figure 1b), a crowd passes beneath a (near-)stationary camera, allowing for the temporal reconstruction of the trajectories of all individuals within the FoV. This configuration produces short trajectories of a continuous crowd over a fixed area. In the present study, the stationary camera is positioned above a bridge deck.

Based on these distinctions, the measurements are classified into: (1) group events, in which a fixed-size group is tracked from a moving camera position, and (2) continuous stream events, in which an unbounded crowd is tracked from a stationary camera position.

#### 2.1.1. Fixed-Size Group Events

During the group events, a fixed number of runners ran on an athletics track at a common running velocity and an arbitrary group arrangement. The athletics track offers several advantages over open, unstructured environments, including the possibility of performing multiple laps. Under controlled experimental conditions with a limited number of participants, each individual was equipped with a coloured hat to enable straightforward person detection, as detailed in the following section. Additionally, runners are instrumented with motion capture sensors to obtain supplementary body motion data.

#### 2.1.2. Large-Scale Stream Events

Additionally, measurements were conducted during two large-scale urban running events with permission from the organizers and informed consent from all participants. The first field study took place during a (half) marathon, with approximately 12,500 participants, while the second involved a 10 mile race, with roughly 6200 participants. In both cases, a footbridge included in the official route of the race was selected as the suitable location to collect the running trajectories. A camera was positioned approximately 30 m above the highest point of each bridge deck. Given the camera characteristics, this setup provided a FoV of about 40 m × 22 m, partially covering the bridge deck, as illustrated in Figure 1b.

### 2.2. Materials

The measurements required a minimal setup. To register the running events, an optical sensor (RGB camera) was used. To enable the localization of the camera in a world coordinate frame, an RTK GNSS receiver was used to measure well-defined reference points in the world space. During the controlled group events, all runners were equipped with coloured hats.

#### 2.2.1. Optical Sensors

A single RGB camera mounted on an Unmanned Aerial Vehicle (UAV) provides a more practical approach compared to a network of fixed cameras, for instance, by eliminating the need for constructing truss structures to position cameras at the heights required for aerial data collection. In the present study, two different types of quadcopters (DJI, Shenzhen, China) were used to capture the required aerial video footage, as shown in Figure 2. The DJI Mini 3 Pro (Figure 2a) is a lightweight UAV of less than 0.25 kg equipped with a high-resolution camera and a fully stabilized tri-axial gimbal, offering ease of deployment without comprising on video quality. The camera has a 1/1.3 inch CMOS sensor and is able to shoot 4K video. The UAV can reach a flying speed up to 16 m/s and has a maximum flight time of 30 min [32]. Alternatively, the DJI Mavic 3 Enterprise (Figure 2b) provides enhanced imaging capabilities, including a larger 4/3 inch CMOS sensor and extended flight time, making it ideal for covering larger areas or conducting prolonged measurements [33]. Nonetheless, its total weight of nearly 1 kg makes it subject to stricter regulatory requirements. A notable advantage of the Mavic 3 Enterprise is its compatibility with a Real Time Kinematic Global Navigation Satellite System (RTK GNSS) module, which enables real-time corrections for centimeter-level camera positioning accuracy.

Due to regulatory and practical considerations, the DJI Mini 3 Pro was used during the stream events, where its lightweight design and flexibility proved advantageous. Meanwhile, the DJI Mavic 3 Enterprise was deployed for the group events on the athletics tracks, where the use of an RTK GNSS module was crucial.

#### 2.2.2. RTK GNSS Receiver

Ground Control Points (GCPs) are an integral part of the trajectory reconstruction methodology, as the points are used at various processing stages such as camera calibration, pose estimation and geo-referencing of point clouds. A GCP is a well-defined location within a scene, typically obtained through precise surveying or GPS measurements, and is selected from easily identifiable permanent scene features, such as road markings or corners. Temporary, self-defined markers can also be used when permanent features are unavailable.

In this study, a combination of permanent and temporary features is used to ensure that at least 20 markers were uniformly distributed across the scene for robust analyses. On the athletics tracks, lane markings served as permanent GCPs (Figure 3a,b). Conversely, on the featureless footbridge deck, temporary markers were created (Figure 3c). All GCPs were measured using the handheld RTK GNSS receiver Emlid Reach RX [34] (Emlid Tech Korlátolt, Budapest, Hungary), which offers horizontal and vertical positioning precision of 7 mm + 1 ppm and 14 mm + 1 ppm, respectively. To prevent signal obstruction, measurements were taken with the receiver mounted on a 1.8 m stick.

#### 2.2.3. Coloured Hats

When participant instrumentation was feasible, each person was provided with a coloured hat, serving as a key feature in the person detection process. For accurate and robust detections, the hat colour needs to be distinctly separable from the background under varying acquisition conditions.

To identify the optimal hat colour for the intended measurements, candidate colours were evaluated under both shaded and sunlit conditions. The analysis was conducted in the CIELAB colour space, following the approach presented in [31]. The candidate colours providing the strongest contrast with the characteristic reddish tone of the athletics track were blue, magenta and dark green. However, because blue is common in running attire and green increases the risk of false positives from vegetation along the track, magenta was chosen as the optimal hat colour.

### 2.3. General Computer Vision and Photogrammetry Techniques

The proposed trajectory reconstruction framework, detailed in Section 3, combines techniques from computer vision and photogrammetry. Two detection strategies are applied: colour-based image segmentation for small groups and an open-set object detector for larger crowds. The reconstruction of world space and camera parameters is achieved through Structure-from-Motion, which relies on solving the Perspective-n-Point problem. Image coordinates are then projected to world space using a homography.

#### 2.3.1. Open-Set Object Detector

When distinctive colour features are unavailable for object detection, alternative strategies are required. Instead of colour-based foreground extraction, background subtraction can been used to isolate moving objects [35,36,37]. While effective in controlled settings, these methods are sensitive to background quality and often fail to generalize. For the present application, a feature-based strategy using an open-set object detector is therefore adopted.

Traditional deep learning detectors, including single-stage, two-stage and transformer-based detectors, typically operate in a closed set, limiting their ability to detect novel object categories [38]. Open-set detection addresses this by identifying both known objects and instances outside predefined classes, which is essential in dynamic environments. Furthermore, stronger closed-set detectors generally improve open-set performance [24].

In this work, person detection is performed with the transformer-based open-set detector *Grounding DINO* [24], which detects arbitrary objects from textual prompts. Its architecture captures global image context, enabling robust detection in challenging scenarios such as overhead views, crowded scenes and partial occlusions, making it well suited for the present application.

#### 2.3.2. Structure-from-Motion

Structure-from-Motion (SfM) reconstructs 3D scenes from 2D images while simultaneously estimating the camera geometry. In this way, the camera poses are relatively aligned with a detailed 3D photogrammetric model of the scene. The method has seen extensive development focused on improving accuracy, robustness and scalability [39,40,41]. For a more comprehensive overview of SfM methodologies and applications, the reader is referred to [42,43]. A typical SfM workflow contains at least the following steps:Features such as distinctive pixels (keypoints), patches and image descriptors are extracted from each image and formulated mathematically, e.g., using the Scale-Invariant Feature Transform (SIFT) algorithm [44].Features identified across multiple viewpoints are matched using, for instance, Approximate Nearest Neighbors (ANN) [45] and the Random Sample Consensus (RANSAC) iterative method [46]. This method ensures robust feature matching by eliminating outlier correspondences. For instance, transient features like pedestrians or moving objects in the scene are automatically excluded from the matched images and subsequent image processing [42].The image correspondences impose constraints on the position and orientation of the moving camera. Following a thorough assessment of matched features, the predominant transformation between two viewpoints is determined using principles of epipolar geometry. This results in the construction of the fundamental matrix, or in the case of a calibrated camera, the essential matrix, which relates the poses of both cameras [47].Camera poses and the 3D positions of matched features are estimated through aerial triangulation. Since photogrammetry inherently relies on a scaling factor, the 3D reconstruction should be appropriately scaled and, if necessary, geo-referenced. This can be achieved by incorporating scaling elements, GCPs or geospatial data obtained from a UAV equipped with a Real-Time Kinematic (RTK) GNSS receiver.Initial estimates are refined through iterative bundle block adjustment, and the 3D reconstruction of the scene can be enhanced using techniques such as dense stereo matching.

#### 2.3.3. Perspective-n-Point

The Perspective-n-Point (PnP) problem involves estimating the pose of a calibrated camera given a set of correspondences between known 3D points in world space and their 2D projections in the image. PnP is a fundamental component in many computer vision and photogrammetry pipelines, including SfM, enabling accurate camera localization and scene reconstruction. Numerous algorithms have been proposed to solve PnP efficiently and robustly, balancing computational complexity and accuracy [48]. For a comprehensive review of PnP methods and their applications, the reader is referred to [48,49]. In the present study, PnP is explicitly solved in events recorded from a stationary perspective, where consecutive frames lack sufficient parallax for the SfM algorithm to reliably estimate the camera pose.

#### 2.3.4. Homography

A planar homography is a projective transformation that relates points between two planes with a known relative orientation. In the present application, it is used to map 2D homogeneous points in the image plane, $m={\left[x,y,w\right]}^{T}$, to 3D homogeneous world coordinates, $M={\left[X,Y,Z,W\right]}^{T}$. The world coordinates *M* are constrained to the homography-plane π, described in homogeneous coordinates by:(1)$$A_{\pi }X+B_{\pi }Y+C_{\pi }Z+D_{\pi }W=0$$
where $A_{\pi }$, $B_{\pi }$, $C_{\pi }$ and $D_{\pi }$ are respectively the plane coefficients, with $C_{\pi }$ different from zero. Restricting the projected world coordinates to this homography-plane allows to construct the homography matrix $H_{\pi }$ which maps the image coordinates *m* to world coordinates $\overset{˘}{M}={\left[X,Y,W\right]}^{T}$:(2)$$m=\left[\begin{array}{c} x\\ y\\ w \end{array}\right]≃H_{\pi }\left[\begin{array}{c} X\\ Y\\ W \end{array}\right]=\left[\begin{array}{ccc} h_{11} & h_{12} & h_{13}\\ h_{21} & h_{22} & h_{23}\\ h_{31} & h_{32} & h_{33} \end{array}\right]\left[\begin{array}{c} X\\ Y\\ W \end{array}\right]$$
with ≃ denoting equality up to a non-zero scale factor. $h_{ij}$ is the element in the *i*th row and *j*th column of the homography matrix $H_{\pi }$, whose exact form depends on the adopted projection model. The undistorted image coordinates *m* can then be mapped to world coordinates by:(3)$$\left[\begin{array}{c} X/W\\ Y/W \end{array}\right]=\left[\begin{array}{c} \frac{H_{\pi 1}^{-1}{\left[x,y,w\right]}^{T}}{H_{\pi 3}^{-1}{\left[x,y,w\right]}^{T}}\\ \frac{H_{\pi 2}^{-1}{\left[x,y,w\right]}^{T}}{H_{\pi 3}^{-1}{\left[x,y,w\right]}^{T}} \end{array}\right]$$
with $H_{\pi i}^{-1}$ the *i*th row of the inverse matrix of $H_{\pi }$.

## 3. Vision-Based Trajectory Reconstruction

The trajectory reconstruction process is divided into four main steps: object detection (Section 3.1), object tracking (Section 3.2), trajectory reconstruction using perspective projection with a planar homography (Section 3.3) and trajectory refinement via a Kalman filter (Section 3.4). Since group and stream events are recorded under different acquisition conditions, two distinct processing pipelines are developed and schematically summarized in Figure 4. The workflow highlights distinctions based on static versus dynamic camera perspectives and the feasibility of participant instrumentation, with primary differences occurring in the detection and tracking stages and in the determination of camera and scene parameters.

The first pipeline uses colour-based image segmentation to detect runners, facilitated by participant instrumentation, and subsequently associates predicted and observed colour blob locations from a moving camera perspective. The second pipeline, which uses an open-set object detector to identify and localize runners, initializes feature-based tracking from the resulting bounding boxes. The camera maintains it position throughout the measurements. Although the detection and tracking steps are computationally more intensive, this method provides broader generalization across datasets.

Both pipelines apply perspective projection with a planar homography to project the trajectories to world space. The trajectories are then refined using a Kalman filter and Rauch–Tung–Striebel (RTS) smoother [50] to mitigate random process noise.

A key consideration in the reconstruction method is the use of a homography-plane, defined at a fixed vertical offset from the running surface. While the details are provided in Section 3.3, it is important to state upfront that the method relies on tracking the centre of each runner’s head, rather than other body parts, such as the lower extremities.

### 3.1. Overhead Person Detection

A first step towards reconstructing the running paths is the identification of the runners in each frame. Selecting a suitable detection method is critical, as it directly influences subsequent processing steps. In this context, two overhead person detection approaches are presented. The first one relies on participant instrumentation, allowing to isolate the objects of interest in the frame based on distinct colour features. Secondly, a more general but computationally extensive approach performs object detection by using an open-set object detector. Each approach starts with the raw video frames (no image pre-processing required).

#### 3.1.1. Colour-Based Image Segmentation

The foreground, being the persons, is easily extracted from the background, which includes everything else in the frame due to the instrumentation of each runner with a magenta coloured hat. For this, the RGB frames were first converted into a binary representation using a constructed binary colourmap $B_{c}$. Each pixel is in this way classified as belonging to either the foreground, based on its match with the selected colourmap $B_{c}$, or the background. This is executed in Matlab using the built-in functions im2double (conversion to double precision) and rgb2ind (conversion to indexed image using the binary colourmap and minimum variance quantization), under default settings.

The colourmap is constructed from 50 frames, randomly selected from the video frames. Each frame, represented in the RGB space, is converted to an indexed image using a colourmap of 500 quantized colours, obtained through minimum variance quantization. Next, the colourmap is mapped in the CIELAB colour space and k-means clustering is applied to segment the colours into 15 clusters, minimizing the sum of squared Euclidean distances, thereby producing spherical clusters. The colour clusters corresponding to magenta shades are retained to generate a binary colour map.

Following image segmentation, morphological image processing operations are applied to each frame to suppress noise, smooth detection boundaries and improve the subsequent blob analysis accuracy. This is implemented in MATLAB R2023a [51] using the Image Processing Toolbox (version 11.7). The procedure starts by filling circular objects with a radius between 5 and 50 pix (fillSphericalObjects). Followed by the steps: horizontal line are removed (morphological structuring element strel with length 5 and imerode), image closing (imclose with disk size 3) and image dilation (imdilate with disk size 2).

Foreground regions, or blobs, are then identified using a blob analysis, performed in MATLAB using the blob analysis object (MATLAB BlobAnalysis system object, Computer Vision Toolbox version 10.4). This basic object localization step determines the centroids of each detected person’s head (i.e., hat) in each frame. Figure 5 illustrates the results of this detection approach, showing the identified centroids of the detected runners from Figure 5a in Figure 5b.

#### 3.1.2. Object Detection Using an Open-Set Object Detector

For events without participant instrumentation, person detection is performed using the pre-trained open-set object detector Grounding DINO [24]. The textual prompt *person* was chosen, as it provided the most reliable results. Textual and bounding-box confidence thresholds were set to 0.1 and 0.3, respectively, prioritizing the detection rate and reducing false negatives. This is particularly important for small, top-view runners. Downstream filtering and tracking mitigate any resulting false positives. The network outputs both the coordinates of localized objects (rectangular bounding boxes) and the associated confidence scores.

To ensure applicability in the present context, two additional considerations are made:To reduce the computational cost associated with detecting and processing numerous bounding boxes, Grounding DINO was applied exclusively within a pre-defined Region of Interest (RoI). The rectangular RoI was positioned along one of the vertical edges of the frame, where runners first appeared, as shown on Figure 6a.Since the subsequent tracking step relies on these bounding boxes as input, a filtering step was applied prior to initiating tracking. This multi-step procedure uses Non-Maximum Suppression (NMS) and a bounding box similarity score (≥0.65), based on the total and matched image features of the image patched in two bounding boxes, to remove redundant boxes. The strongest bounding boxes are selected using the MATLAB built-in function selectStrongestBbox, which uses the confidence scores as provided per bounding box by Grounding DINO. Moreover, it was observed that shadows often caused false positives. These were mitigated by applying aspect ratio constraints and comparing the colour space against a set of template shadow images. In this way, redundant detections are excluded, overlapping bounding boxes of partially detected runners are merged, bounding boxes containing multiple runners are separated and false positives caused by shadows are removed. While this filtering process applies generic techniques, certain steps may require tuning under different acquisition conditions.

The results are illustrated in Figure 6, where Figure 6a,b display the detector successfully identifying all the runners in the RoI, although multiple shadow detections are included. Figure 6c demonstrates the outcome after processing, with only the true runner detections retained. After the pre-processing, 43% and 84% of the initial detections were retained for the marathon and 10 mile event, respectively. The higher proportion of removed bounding boxes in the first urban running event can be attributed to a significantly greater number of false positives caused by shadows.

### 3.2. Overhead Person Tracking

The second processing step requires Multi-Object Tracking (MOT), which is a key task in many computer vision applications and involves the tracking of objects within a scene while maintaining a unique identifier for each detected object. The primary objective is to reconstruct the trajectories of multiple objects, either in real-time or offline, across a sequence of images [52].

Since online tracking is not required within this framework, the position of an object can be inferred from past, present and future localized objects [53]. Tracking relies on the output of the person detection step and is consequently dependent on the specific applied approach. During the group events, runners were tracked continuously by a moving camera, navigating through an open world space. Tracking begins from the first frame, associating each runner with a unique identifier, and continues until the final frame. This method relies on the location of identified blobs in the each frame, treating the task as an object association problem (see Section 3.2.1). Instead, during the stream events, a stationary camera was used while a crowd moved underneath. Feature-based tracking is initiated when a runner is detected and ends when the runner exits the FoV. This approach tracks identified image features by matching them across frames (see Section 3.2.2). Both methods result in the reconstruction of each runner’s trajectory in the image plane.

#### 3.2.1. Blob Association

From colour-based image segmentation, highly accurate object localization is achieved and does not strictly require filtering at this stage. Consequently, the raw blob locations are processed in the association method.

To avoid the complexities associated with 3D scene reconstruction and errors introduced during the projection step, the tracking procedure operates entirely in image coordinates. However, as a consequence, the relative motion of the camera w.r.t. the objects in the scene becomes an additional factor to consider during the association process.

Association of the unique runners to the identified blob locations is initiated using their starting positions in the first frame. In this way, an unique identifier is assigned to each blob. For every subsequent frame, the identifiers are used to associate the identified blobs in the current frame to the ones of the previous frame. Association of blob *i* in the current frame with blob *j* in the previous frame is achieved by minimizing the distance of the identified and predicted centroid:(4)$$\begin{array}{c} d_{ij}=\min\left|\left|b_{i}^{c}-\left(b_{j}^{p}+v_{j}\right)\right|\right| \end{array}$$
with $d_{ij}$ the minimal euclidean distance in the image plane while $b^{c}$ and $b^{p}$ are the coordinates of the centroids in the current and previous frame, respectively. To predict the locations of the centroids in the current frame, a movement vector v is defined, accounting for the relative movement of the associated blobs between the previous and the second previous frame, relative to the camera motion.

#### 3.2.2. Feature-Based Tracking

The filtered output of the open-set object detector, i.e., bounding boxes, are used to initiate a feature-based tracking procedure for each uniquely identified runner. In this work, the KLT tracker [54] tracks image KAZE features across consecutive frames, which are identified using (detectKAZEFeatures) applied to the image within the bounding box. The MATLAB PointTracker system object with its default internal configuration is used. Key parameters, including the number of pyramid levels (3), window size (50 pix), termination criteria and a minimum of 5 valid matched features, followed the built-in KLT implementation of MATLAB without further required tuning. The initial features are tracked, and the centroid is updated by repositioning it to the centre of the image patch with the highest density of tracked features. If no features (at least 5 valid matched features) can be matched and the object is not near the frame edges, the tracker re-initializes using features from the previous frame at the tracked location. Tracking is terminated if the runner crosses an image boundary.

The tracking procedure begins with a forward pass, processing frames in ascending order, starting from the bounding boxes defined within the RoI located along one of the vertical edges of the frame. Since the methodology assumes that the runner’s head is tracked, features from the head region should ideally be used. However, at the start of the tracking procedure, the head is generally not accurately localized, as the detection step does not consider this. By applying a feature-based tracking approach, the head location is gradually approximated by automatically shifting the tracking focus towards the head, as the head represents a stable feature region across the frames. Consequently, the largest errors occur at the beginning and decrease as the runner progresses towards the image centre. To refine the trajectories, the initial forward tracking continues until the runner crosses the image center. At this moment, the forward tracking has approximated the head and its associated features. Subsequently, both forward and backward tracking procedures are initiated, tracking the runner from the image center towards the vertical frame edges.

The method is illustrated in Figure 7 and Figure 8. Figure 7 presents the initial forward tracking and the results after applying the refined procedure, which combines forward and backward tracking. Figure 8 shows the total trajectory of the runner in Figure 7, in image coordinates. The improvement is particularly evident at the start (see Figure 7c and Figure 8), where the refined tracking procedure closely approximates the head location, whereas the initial tracking does not.

### 3.3. Perspective Projection with a Planar Homography

Once the trajectories are obtained in the image plane, perspective projection is applied to map the image coordinates to world coordinates using the pinhole camera model [55]. However, since a single camera is used to record the events, depth information is inherently lost. To resolve this ambiguity, a planar homography is used, which requires reconstructing the running surface (athletic track or bridge deck) to define a homography-plane in 3D space. Additionally, both intrinsic and extrinsic camera parameters must be determined.

#### 3.3.1. Pinhole Camera Model

The standard linear pinhole camera model is represented by a projection matrix *P*, which maps a 3D world coordinate $M\;=\;{\left[X,Y,Z,W\right]}^{T}$ to a 2D (undistorted) image coordinate $m\;=\;{\left[x,y,w\right]}^{T}$, both expressed in homogeneous coordinates. To preserve the perspective relations in the scene, when using homogeneous coordinates, the general mapping of the pinhole camera model is equal ‘up to a non-zero scaling factor’ and reads [55]:(5)$$m≃PM=K_{c}\left[\;\;R^{T}\;\;\;|\;\;-R^{T}t\;\;\right]M$$
with $K_{c}$, *R* and *t* the calibration matrix, the rotation matrix and the translation matrix of the camera, respectively. The calibration matrix $K_{c}$ is a 3 × 3 matrix, the projection matrix *P* is a 3 × 4 matrix and | represents the matrix concatenation. The matrix $K_{c}$ includes the focal lengths $f_{x}$ and $f_{y}$, and the coordinates of the principal point ($p_{x}$, $p_{y}$), expressed in pix.

The extrinsic parameters of the camera are represented by the rotation matrix *R*, describing the orientation of the camera relative to the world coordinate system, and the translation matrix *t*, positioning the origin of the camera in the world coordinate system. *R* is a 3 × 3 matrix, fully defined by its three Euler angles E={ϕ,θ,ψ} and $t={\left[t_{X},t_{Y},t_{Z}\right]}^{T}$ is a 3 × 1 vector. The translation captures the flight height of the UAV, which is in this study targeted to be 30 m.

A key limitation of the ideal pinhole camera model is its neglect of lens distortion, which results from optical aberrations due to imperfections or misalignments in the lens system, such as manufacturing defects. Lens distortion is typically decomposed into radial and tangential components. The widely used Brown–Conrady model [56,57] accounts for these distortions through a polynomial formulation. The model defines a non-linear mapping between the distorted image coordinates $m_{d}$, corresponding to the observed pixel locations, and the undistorted coordinates $m_{u}$ that an ideal pinhole camera would project. Consequently, the overall mapping from world coordinates M to distorted image coordinates $m_{d}$ becomes inherently non-linear:(6)$$m_{d}=K_{c}D\left(M\right)$$

The distortion operator D uses the equations proposed by the Brown-Conrady lens distortion model [56,57].

#### 3.3.2. Camera Parameter Estimation from a Moving Camera Perspective

Both intrinsic and extrinsic parameters are jointly estimated using SfM [58] via the software package Metashape by Agisoft [59] (Professional 1.5.0 build 7011). Metashape’s default parameters are used unless explicitly noted in the paper (alignment and reconstruction parameters quality: medium; key point limit: 40,000; tie point limit: 4000; depth filtering: aggressive; point-based aligned chunks, image matching quality: medium).

The process begins with the application of the SfM algorithm to a set of source images of the athletics track, resulting in a photogrammetric reference model of the 3D scene. These images are captured using a UAV equipped with an RTK GNSS receiver, while GCPs are measured using a handheld RTK GNSS receiver. This process yields a highly accurate, geo-referenced photogrammetric model together with a triangulated tie-point cloud of the 3D scene. The frames capturing the running events are subsequently aligned with the reference model through bundle adjustment, resulting in refined camera parameters.

SfM reconstruction and bundle adjustment were performed using Metashape’s default parameters unless explicitly noted in the manuscript (alignment and reconstruction parameters quality: medium; key point limit: 40,000; tie point limit: 4000; depth filtering: aggressive; point-based aligned chunks, image matching quality: medium).

The outcome of this procedure is visualized in Figure 9 and Figure 10 for one representative event. Figure 9 displays the resulting photogrammetric model alongside the aligned camera positions, while Figure 10 presents the camera’s translation in world coordinates over time in the horizontal plane (Figure 10a), the flight altitude relative to the average terrain elevation (Figure 10b), and the deviation in orientation from an ideal nadir configuration, defined as $E_{\mathrm{NADIR}}=\left\{0,0,90\right\}$ deg (Figure 10c).

#### 3.3.3. Camera Parameter Estimation from a Stationary Camera Perspective

A near-stationary camera perspective introduces minimal parallax between frames, producing an almost degenerate image set for SfM and limiting the reliability of 3D reconstruction and camera pose estimation. To overcome this, multiple GCPs were placed within the scene (Figure 11a), enabling pose estimation via the PnP algorithm using 3D–2D image correspondences. In this way, the frames are processed individually rather than collectively as in the SfM approach. Prior to this, intrinsic camera parameters were determined through checkerboard calibration (square size: 0.039 m), yielding a mean re-projection error of 0.23 pix (Figure 11b), confirming the calibration accuracy for subsequent processing.

To avoid solving PnP for a significant number of frames, the raw set of video frames is first decimated into a smaller batch. This decimation is feasible due to the (near-)stationary camera position. For this, video stabilization is applied to identify a reduced set of key event frames for which the PnP problem is solved. The stabilization is achieved by calculating the 2D geometric (projective) transformation between frames based on matching feature pairs. An initial transformation of each frame with a reference frame is computed and further optimized using KLT-tracked features across both frames.

The approach works by designating the first frame as the reference frame, with sequential transformations estimated for all subsequent frames. The reference frame is updated when either (1) the number of matched features drops below a predefined threshold or (2) the transformation matrix becomes nearly singular. These transformations are computed using MATLAB’s built-in function estgeotform2d from the Computer Vision Toolbox [51]. For the subset of decimated frames, the PnP problem is solved using estworldpose, which implements the perspective-three-point (P3P) algorithm. Outlier correspondences are removed using the M-estimator Sample Consensus (MSAC) algorithm, and camera poses are estimated. The built-in MATLAB function requires a calibrated camera and can incorporate lens distortion parameters. In the final step, both 3D points and camera poses are refined through bundle adjustment using the bundleAdjustment function. This procedure minimizes re-projection errors via a variant of the Levenberg–Marquardt algorithm. The resulting re-projection errors are evaluated in Section 4.1.1.

#### 3.3.4. Homography-Based Projection

Even with known camera intrinsic and extrinsic parameters, re-projecting image detections to world coordinates is limited by depth ambiguity. Depth ambiguity arises because a 2D image inherently loses the depth information present in a 3D scene. Distances are typically recovered using stereo vision, where two cameras capture the scene from slightly different viewpoints, allowing corresponding points to reduce the ambiguity.

Depth maps can mitigate depth ambiguity in monocular vision by encoding the distance from the camera to each scene point. They can be obtained through 3D scanning or reconstructed via photogrammetry, such as SfM, allowing ray-casting by projecting rays through the image plane and intersecting them with objects based on depth values. However, photogrammetric models typically represent only static elements, as dynamic objects are excluded from feature matching, making this approach unsuitable for the present application. Instead, a homography-based projection is applied, which is less sensitive to reconstruction errors, computationally efficient and independent of moving object feature matching.

In Section 2.3.4, the basic principles of a homography were introduced. The homography-plane π, orientated in world space, constrains the possible positions of a runner’s head and is defined at a fixed vertical offset from the ground, corresponding to the runner’s height. This highlights the importance of tracking the head rather than other body parts. Combining Equation (Equation 5) with Equation (Equation 2) yields the homography coefficients:(7)$$\begin{array}{cccccc} h_{i1} & =p_{i1}-\frac{A_{\pi }}{C_{\pi }}p_{i3}, & h_{i2} & =p_{i2}-\frac{B_{\pi }}{C_{\pi }}p_{i3}, & h_{i3} & =p_{i4}-\frac{D_{\pi }}{C_{\pi }}p_{i3} \end{array}$$
where $h_{ij}$ and $p_{ij}$ denote the elements in the *i*th row and *j*th column of the homography matrix $H_{\pi }$ and the camera projection matrix P, respectively.

The homography concept is illustrated in Figure 12a, where an aerial image captures a 3D scene, runners are detected in the image plane and a homography-plane is defined to project their positions. The definition of this latter plane varies depending on the processing pipeline and measurement conditions:For the group events, the triangulated point cloud of the athletics track, obtained via the SfM algorithm, is used. However, only a limited area around a runner is considered when modelling the homography-plane. The area, defined in the image plane, corresponds to a square of 2 m × 2 m. Figure 12b conceptualizes the approach. In Section 4.1.2, the influence of using a (small) homography-plane with a fixed offset above the ground is investigated.For the stream events, involving a near-stationary camera hovering above a footbridge, the geometry of the bridge deck is used to reconstruct the homography-planes. When the height and gender of the runner is unknown, it is suggested to use a fixed offset of 1.7 m between the ground and the homography-plane [60].

### 3.4. Trajectory Refinement

To effectively account for random measurement errors, a Kalman filter and RTS smoother are applied, using a constant velocity motion model. The RTS smoother uses the forward Kalman estimates and the same state transition and process noise covariance matrices to compute smoothed states.

#### 3.4.1. Measurement Noise

To quantify the maximum random measurement error, an evenly spaced crowd, represented by a grid of points each separated by 1 m, was simulated on a horizontal plane. A camera object, configured with intrinsic parameters obtained through (self-) calibration, captured the scene from a flight altitude of 30 m, relative to the plane. The true locations of the grid points were transformed into image coordinates using perspective projection (true detections), covering the full FoV. The uncertainty associated with each grid point is subsequently modelled by a bivariate normal distribution. The uncertainty is then propagated back to world coordinates to calculate the maximum error. The latter is defined as the maximum distance between the true coordinate and the 95% confidence line. Variances are assumed constant across the image plane and equal in both horizontal and vertical directions to 4 pix^2^ ($\sigma_{h}^{2}=\sigma_{v}^{2}$), with no covariances. This assumption is supported by the consistent size of detected blobs, segmented via colour-based image processing, given a flight altitude of 30 m, and further confirmed by visual inspection of head sizes within frames from the stream events.

To propagate the measurement error to world coordinates, the bivariate normal distribution, defined by the undistorted image coordinates m, is transformed using Equation (Equation 3). The probability density function is scaled by the determinant of the Jacobian, and an eigenvalue decomposition yields the maximum horizontal measurement error.

As the random measurement error in the image plane is identical for both pipelines, the absolute error in world coordinates depends solely on the camera. For both UAVs, this error remains below 0.1 m across the image plane.

#### 3.4.2. Process Noise

Process noise accounts for modelling approximations and is interpreted, in the constant-velocity model, as a piecewise constant acceleration modelled by zero-mean white Gaussian noise. Rather than assigning arbitrary variances, the unknown process-noise covariance matrix *Q* was estimated using the Expectation–Maximization (EM) algorithm [61]. The EM procedure was initialized with a diagonal covariance matrix whose positional and velocity variances were set to (0.5 m)2 and (0.25 m/s)2, respectively. In each iteration, the Kalman filter and RTS smoother produced state estimates ${\hat{s}}_{\left[1:N|N\right]}$, after which the closed-form update rule from [61] was applied to compute an improved estimate of *Q*. The iterations continued until convergence of the process-noise covariance matrix was achieved, which was in all cases below 10 iterations. This data-driven procedure ensures that the process noise reflects the observed motion dynamics rather than user-specified assumptions.

## 4. Results and Discussion

### 4.1. Evaluation of the Systematic Measurement Errors

The margin of error of the reconstructed trajectories is assessed by evaluating the impact of the most significant systematic measurement errors. The primary error sources are: (1) re-projection errors due to imperfections in camera calibration, (2) inaccuracies arising from the use of a homography, and (3) when using an open-set object detector an additional detection error might be introduced. Lens distortions are considered in the calibration re-projection error, while the other analyses are applied directly using Equation (Equation 5) directly.

#### 4.1.1. Imperfect Camera Calibration

The systematic error resulting from imperfect camera calibration was assessed using the re-projection errors of the GCPs. This error arises from inaccuracies in estimating both intrinsic and extrinsic parameters, including the camera’s position and orientation. For a moving camera, these parameters are obtained using Agisoft Metashape [59], whereas for a stationary camera, they are derived through direct calibration and solving a PnP problem.

In both processing pipelines, GCPs were collected to improve reconstruction accuracy, assess camera calibration or solve the PnP problem. These GCPs were measured using an RTK GNSS receiver, typically achieving an accuracy of 0.01 m to 0.02 m [34,62]. The re-projection error is determined by comparing the projected positions of the GCPs with their corresponding image coordinates. However, even with optimal calibration, inaccuracies in GCP positioning can contribute to the overall re-projection error. The positional error introduced by the RTK GNSS receiver is estimated at a maximum of 0.03 m on the athletics track and 0.05 m during the large-scale running events, with the difference attributed to the types of markers used. Applying the GSD, these spatial errors correspond to approximately 2.7 pix for the athletics track, 6 pix during the marathon event and 4 pix during the 10 mile event.

Figure 13 presents the average re-projection error per frame. In Figure 13a, 460 frames are randomly selected from all group events, while Figure 13b includes the full set of decimated frames. The average (standard deviation) of the re-projection error of the controlled group events, the marathon event and the stream event are 0.9 pix (0.67 pix), 3.73 pix (0.87 pix) and 3.27 pix (0.94 pix), respectively. The majority of re-projection errors lie well below the maximum expected error under ideal camera calibration conditions. The maximum observed error remains just under 8 pix, corresponding to an absolute positional error of less than 0.1 m. It is noteworthy that in Figure 13a, the GCPs are predominantly located near the image boundaries, where re-projection errors are typically larger. In contrast, runner detections are concentrated near the centre of the frame (e.g., see Figure 1), suggesting that the effective re-projection error in the region of interest is likely well below 0.1 m. Similar patterns are observed in the stationary camera setup, where edge markers contribute most to the error, confirming that the re-projection errors in the central region remain minimal.

Since different processing pipelines and camera setups are applied, a direct comparison of average re-projection errors between the two methods is not straightforward. However, the pipeline involving a moving camera, which relies on the robust SfM procedure and incorporates a high-accuracy photogrammetric reference model with RTK-localized camera positions, is expected to yield the lowest re-projection errors, as observed.

#### 4.1.2. Height Variation of the Homography-Plane

To resolve depth ambiguity, a homography-based projection approach has been proposed. In this method, 2D detections in the image plane are projected on a second homography-plane, positioned with an offset above the ground corresponding to the measured or estimated height of the runner (stand-up posture). However, a fixed offset might introduce systematic errors in the trajectory reconstruction. These errors arise from the natural vertical sway of the head, which varies with their posture during the gait cycle, as well as the simplification of the ground as a perfect planar surface, e.g., due to the vertical curvature of the bridge deck. Moreover, when the person’s height is unknown, it must be estimated.

A virtual scene representative of the measurement setup on the athletics track and the bridge deck is simulated. A grid is defined for each integer image coordinate, effectively placing a virtual runner at every possible location in the frame. These grid points are projected into world space using the homography matrix $H_{\pi }$, as described in Equation (Equation 3). For the athletics track configuration, a planar surface is assumed, whereas for the bridge deck configuration, the actual deck curvature is included. The homography matrix is constructed using the intrinsic parameters, as obtained from [59] or through camera calibration, and a fixed offset from the supporting surface. The virtual camera is positioned at nadir, 30 m above the ground or, for the stream-running events, above the bridge deck. Each grid point is projected twice using two horizontal homography planes separated by a prescribed height difference. The positional difference between both projections quantifies the maximum error introduced by this height difference. Apart from the camera parameters and the assumptions regarding the ground surface, the resulting error depends solely on this height offset.

For the setup involving a moving camera, the absolute error across the image plane for a height variation of 0.1 m is shown in Figure 14a. Notably, the error is zero at the image centre, where the runner aligns perfectly with the camera’s principal axis, and increases radially towards the frame edges. The maximum error for a height variation of 0.1 m is 0.08 m. Restricting the analysis to the central region of the image, where runners are actually detected, reduces the error to 0.042 m. This central region is defined as the rectangle occupying half the width and half the height of the frame.

In addition, Figure 14b presents the maximum error when considering the entire and central part of the image, respectively, at different height variations. When detections are confined to the central part of the image, the maximum error slightly exceeds 0.1 m for a height variation of 0.25 m between the homography-planes. Nevertheless, the occurrence of such height difference between the simulated and true homography-plane is highly unlikely. The vertical sway of the head is likely to be below 0.1 m during running. Moreover, the occurrence of extremely localized bumps or holes in the running path, introducing a height variation of 0.25 m, is virtually non-existent.

Figure 15 presents the results obtained for the grid defined on the footbridges, where the narrow horizontal band corresponds to the bridge deck. The maximum error, approximately 0.07 m, occurs near the frame edge for a height difference of 0.1 m between the homography planes. For participants in the stream events, individual height is estimated using data from the Belgian Health Examination Survey. Assuming a normal distribution with a mean stature of 1.7 m and a standard deviation of 0.1 m, approximately 95% of runners exhibit a systematic height offset of at most 0.2 m. As shown in Figure 14b, this variation produces a maximum absolute positional error of approximately 0.15 m near the vertical frame edges.

#### 4.1.3. Object Detection Error

Finally, the error associated with the person detection is quantified. In the case of colour-based image segmentation, particularly with the accurate head localization provided by magenta-coloured hats, the object detection error is negligible compared to other random and systematic errors. However, when using an open-set object detector, the object detection error becomes the primary factor determining the absolute precision of the reconstructed running paths.

To quantify the systematic detection error, error variances are defined in the image plane and then translated to absolute errors in world coordinates. The detection error is modelled as a bivariate normal distribution, yielding a 95% confidence region that represents the area within which the true head centre is located.

Figure 16 presents the object detection results obtained using an object detector during the stream events. Due to the top view camera perspective and its orientation relative to the unidirectional flow, error variances are defined separately for each image axis, see Figure 16a. The vertical error variance $\sigma_{v}^{2}$ is constant at 10 pix^2^ across the frame. In contrast, the horizontal error variance $\sigma_{h}^{2}$ is 10 pix^2^ at the image centre and increases linearly to 20 pix^2^ towards the vertical edges. This variation accounts for increased head localization uncertainty near the edges in the horizontal direction, while the vertical error remains uniform. No covariance between axes is considered. The circular and elliptical confidence regions near the image center and vertical frame edges, respectively, are shown in Figure 16b–d.

To determine the systematic detection error in world coordinates, a similar approach to that introduced in Section 3.4.1 is used. The PDF resulting from the bivariate normal distribution is appropriately scaled and propagated using Equation (Equation 3). The resulting absolute error in world coordinates resulting from the propagated object detection error in the image plane only changes with the horizontal image coordinate. The maximum errors range from 0.2 m (at the image centre) to 0.5 m (at the image vertical edges).

#### 4.1.4. Combined Systematic Error

For the first processing pipeline, involving a dynamic camera perspective with participant instrumentation, the approach achieves absolute trajectory accuracy of approximately 0.1 m to 0.15 m, accounting for random or systematic measurement errors. In the second pipeline, involving a static camera perspective and detections using an open-set object detector, larger absolute errors are presented, mainly due to the reliance on head localization rather than full-body tracking and the lack of information about the runner’s height. Since the errors do not always accumulate uniformly and are highly dependent on the image coordinates of the detection, the trajectory accuracy is approximated to vary from 0.2 m at the image centre to 0.5 m near the vertical edges. While these errors are systematic and limit the reliability of absolute measurements, the data remain valuable for relative analyses, such as within-subject comparisons. For instance, the approach enables estimates of individual running speeds and provides insights into crowd dynamics by assessing global group densities. Moreover, due to the applied feature-based tracking approach, the relative accuracy is estimated to be significantly higher.

### 4.2. Running Trajectories Characteristics

While a comprehensive analysis of crowd dynamics lies beyond the scope of this study, the reconstructed running trajectories serve as a basis for validating human–human interaction models and examining collective behaviour at both dyadic and group levels under realistic conditions.

#### 4.2.1. Group Events

In the group events, person detection was facilitated by equipping each runner with a magenta coloured hat, allowing the head of each runner to be easily detected and tracked. The high accuracy of the photogrammetric model, implemented in Agisoft Metashape, ensured that the reconstructed trajectories exhibited low absolute positional errors. Consequently, the data are suitable for investigating various group characteristics, such as interpersonal distances between members as a function of running velocity or the impact of changing spatial constrains on the degree of synchronization between the runners.

#### 4.2.2. Stream Events

When the objective is to collect large-scale (big) data for validating crowd dynamics in continuous streams, participant instrumentation is generally unfeasible. In this context, the present work validated the use of an open-set object detector for person detection from a (near-)stationary camera perspective. Following tracking, visual inspection revealed that approximately 2.4% (257) of all runners were missed during the marathon, while 1.4% (150) of all trajectories were discarded due to incorrect or spurious detections. For the second urban running event, 5.3% (178) of all runners were missed, and 1.8% (60) of all tracks were removed. Missing runners were manually added by identifying the runner in a single frame to initialize the tracking procedure.

### 4.3. Suggestions for Future Research

The objective of this work was successfully achieved, providing a practical approach to obtain running trajectories in groups and streams by applying existing detection and tracking models, with minimal fine-tuning to the measurement conditions. In the context of civil engineering and crowd dynamics, the collected datasets facilitate the investigation of the fundamental relationship between density and velocity, as well as the analysis of phenomena such as lane formation within running streams. Moreover, they enable the quantification of collective motion patterns, transient congestion and the probability of occurrence of large streams or discrete group sizes. Consequently, the collection of such data is essential for the development of reliable vibration comfort design guidelines.

Moreover, future work could explore integrating depth information or stereo vision to further improve person detection and trajectory projection accuracy, extend the pipelines to extract additional gait features such as pose estimation from top-down perspectives or oblique viewpoints and evaluate the performance under more diverse environmental conditions. If more enhanced detection and tracking methods become available, the methods could be integrated to streamline certain processing steps, for instance the head localization requirement.

## 5. Conclusions

Two generic image-processing pipelines were developed to reconstruct individual trajectories during realistic in-field activities, such as walking or running. The pipelines are designed to address the specific challenges associated with video footage captured from stationary and moving cameras, applying tailored object detection and tracking strategies for each acquisition scenario. Trajectories are initially extracted in image coordinates and subsequently mapped to world space using perspective projection via a planar homography. Random measurement errors are mitigated through Kalman filtering and smoothing while systematic errors are quantified to evaluate their impact on the trajectory accuracy.

The methodology was applied directly to real-world running data, demonstrating its effectiveness in both group and stream events. While the pipelines are generally applicable, selective fine-tuning at specific stages ensures optimal detection and tracking performance under the given measurement conditions. Overall, the approach provides a robust framework for reconstructing accurate trajectories in diverse in-field scenarios.

The first pipeline is designed for a moving camera position and involves colour-based image segmentation, blob analysis and SfM. This approach achieves absolute trajectory accuracy of approximately 0.1 m to 0.15 m, accounting for random and systematic measurement errors. These precise reconstructions are well-suited for studies of HHI, allowing for a detailed analysis of gait synchronization among neighbouring runners.

The second pipeline, designed for a stationary camera, integrates deep learning-based object detection, feature-based tracking and direct solutions of the PnP problem. This approach produces larger absolute positional errors, primarily due to the object detection step, ranging from approximately 0.2 m at the image centre to 0.5 m near the vertical frame edges. Nevertheless, the data remain valuable for relative analyses, such as evaluating individual-specific parameters (e.g., running velocity) or assessing global crowd densities.

## Figures and Tables

**Figure 1 sensors-25-07577-f001:**
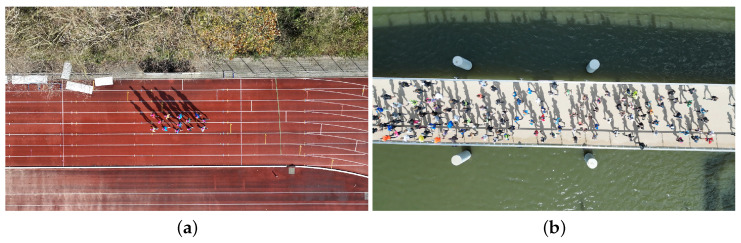
Representative perspectives captured during (**a**) a group event using a moving camera and (**b**) a stream event using a stationary camera setup.

**Figure 2 sensors-25-07577-f002:**
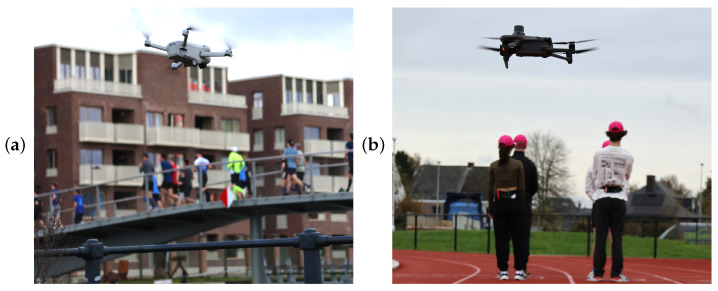
The UAVs used to obtain aerial imagery during operation: (**a**) the DJI Mini 3 Pro and (**b**) the DJI Mavic 3 Enterprise.

**Figure 3 sensors-25-07577-f003:**
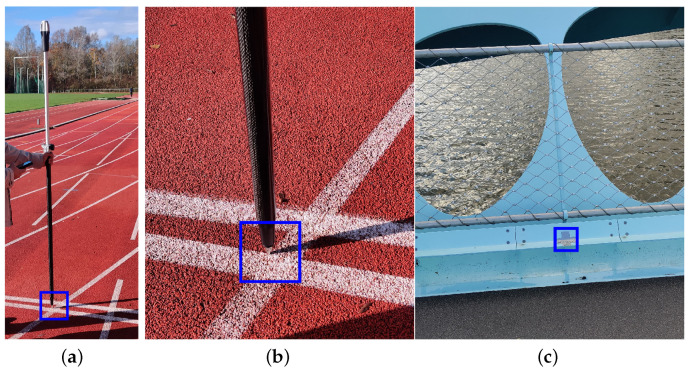
Ground control point (blue box): (**a**,**b**) permanent or (**c**) temporary feature.

**Figure 4 sensors-25-07577-f004:**
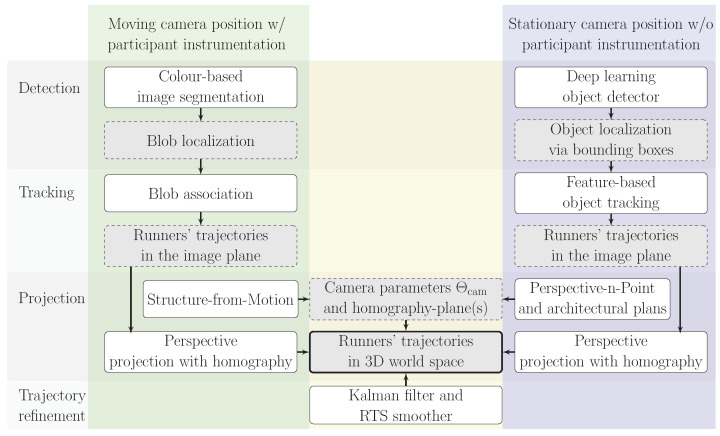
General workflow of Section 3, discussing the vision-based trajectory reconstruction pipeline for a moving (green) or stationary camera (blue) with similar components (yellow). The two independent pipelines are divided in four steps and show the used methods (solid white blocks), the intermediate deliverables (dashed grey blocks) and the final deliverable (solid grey block).

**Figure 5 sensors-25-07577-f005:**
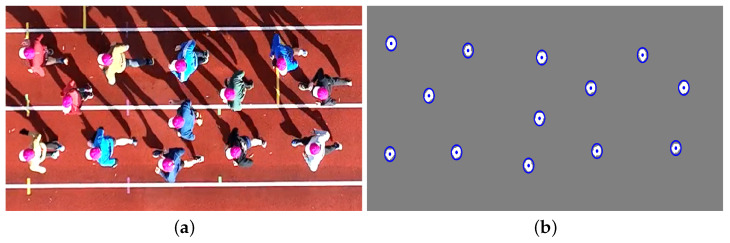
Detection of runners: (**a**) unprocessed frame and (**b**) processed frame using colour-based image segmentation (foreground white, background grey) and blob analysis, with centroids of identified objects (blue markers).

**Figure 6 sensors-25-07577-f006:**
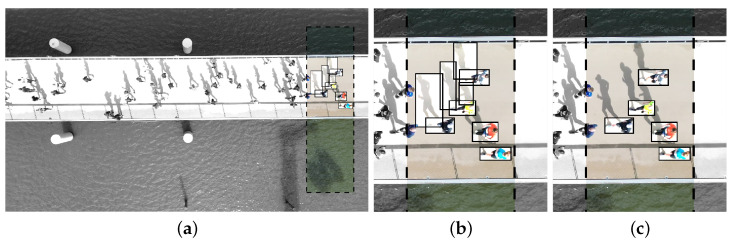
Pre-processing of the open-set object detector outputs: (**a**,**b**) before and (**c**) after removal of redundant and false positive person detections, with (**a**) showing the full frame and (**b**,**c**) a snapshot. The dashed box indicates the RoI for object detection, while the rest of the image is shaded grey. Detections are highlighted as black bounding boxes.

**Figure 7 sensors-25-07577-f007:**
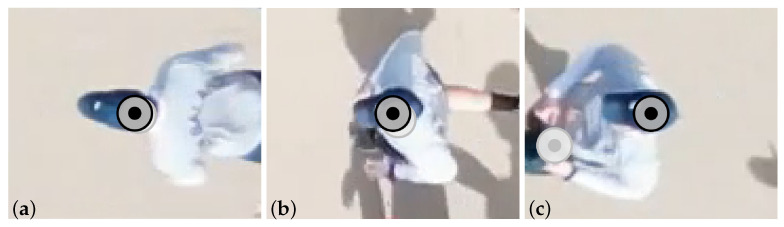
Tracking of the runner’s head: (**a**) near the left edge (exiting the camera FoV), (**b**) at the centre, and (**c**) near the right edge of the full frame (entering the camera FoV). The marker indicates the center of the tracked location: initial forward tracking (grey) and refined tracking procedure with backward and forward tracking from the image center (black).

**Figure 8 sensors-25-07577-f008:**
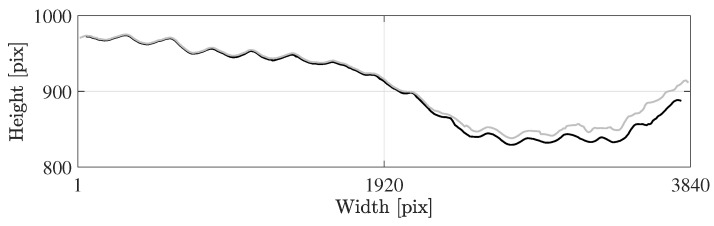
Trajectory of the tracked runner’s head in image coordinates: initial forward tracking (grey) and refined tracking procedure with backward and forward tracking from the image centre (black).

**Figure 9 sensors-25-07577-f009:**
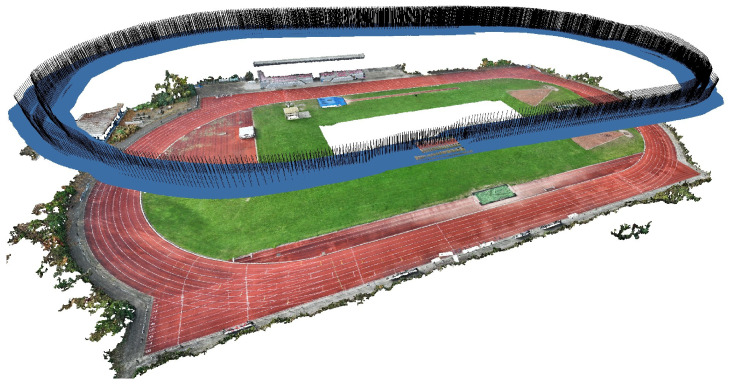
Photogrammetric model as displayed in Agisoft Metashape [59] with cameras (blue image plane and black orientation vector).

**Figure 10 sensors-25-07577-f010:**
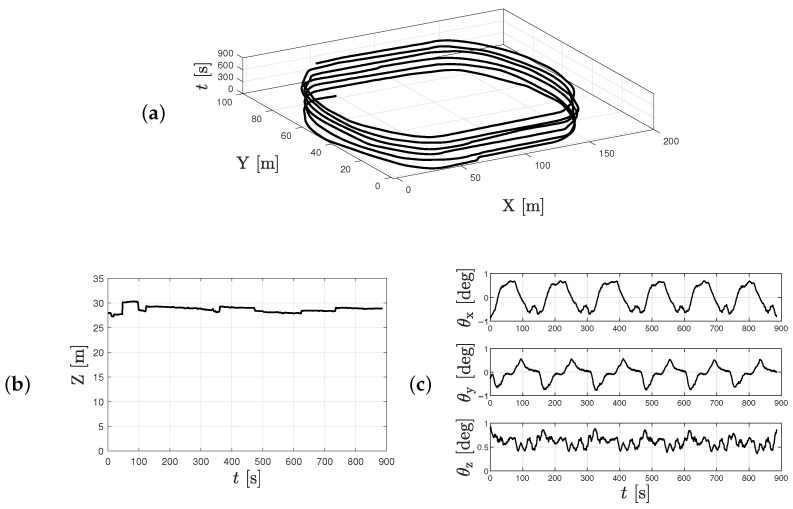
Relative motion of the UAV using SfM for camera parameter estimation: (**a**,**b**) translations along and (**c**) rotation about the *X*, *Y* and *Z* axis.

**Figure 11 sensors-25-07577-f011:**
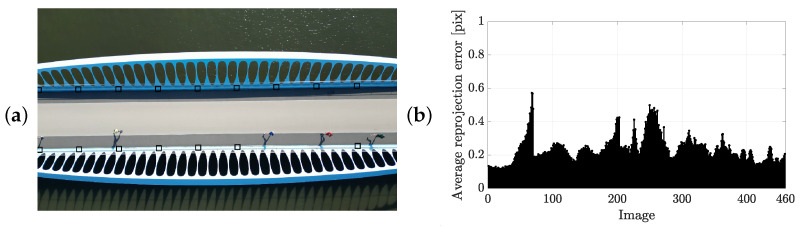
(**a**) Impression of the ground control points (black boxes) used during one of the stream events (10 mile event crossing the Annie Vande Wiele footbridge), viewed from the camera position. (**b**) Camera calibration using a checkerboard.

**Figure 12 sensors-25-07577-f012:**
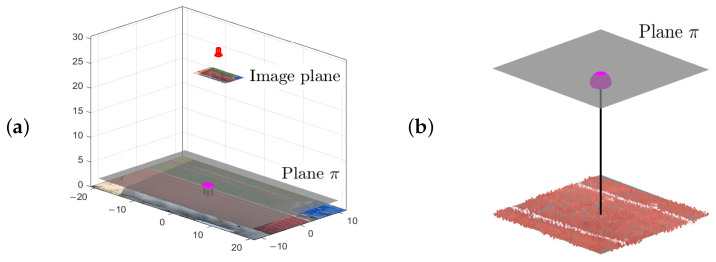
The homography-based projection approach with the homography-plane π (grey): (**a**) aerial perspective from the UAV (red) and (**b**) idealized runner (black) with hat (magenta) intersecting with the plane π at a fixed offset relative to the ground (athletics track point cloud), equal to the runner’s height.

**Figure 13 sensors-25-07577-f013:**
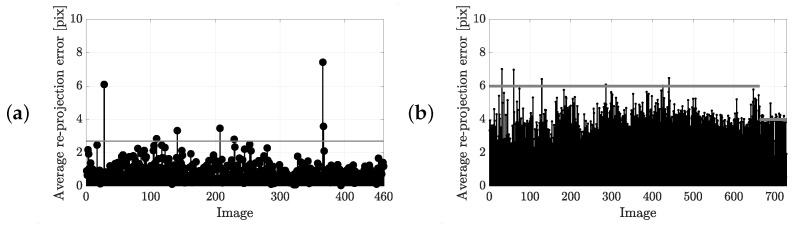
Average re-projection error per frame (black) and the maximum expected error for a perfect camera calibration due to the uncertainty of the used ground control points (grey solid line) for the processing pipeline involving (**a**) a moving and (**b**) a stationary camera.

**Figure 14 sensors-25-07577-f014:**
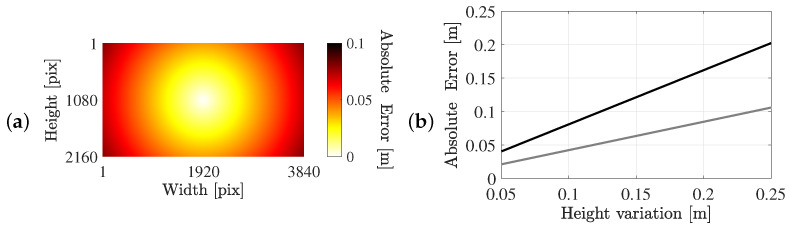
Systematic error introduced due to homography-based projection approach using a fixed offset above the ground. (**a**) The absolute systematic error in world coordinates for a detection in the image plane for an offset of the homography-plane of 0.1 m. (**b**) The absolute systematic error per height variation between the homography-planes, either in the full FoV (black) or the central part of the image (grey).

**Figure 15 sensors-25-07577-f015:**
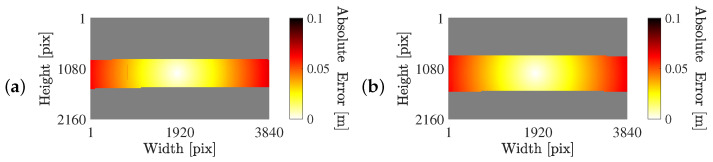
Systematic error due to homography-based projection using a fixed offset of 0.1 m above the ground: (**a**) the Matadi footbridge (marathon event) and (**b**) the Annie Vande Wiele footbridge (10 mile event). The grey space is outside the bridge deck region.

**Figure 16 sensors-25-07577-f016:**
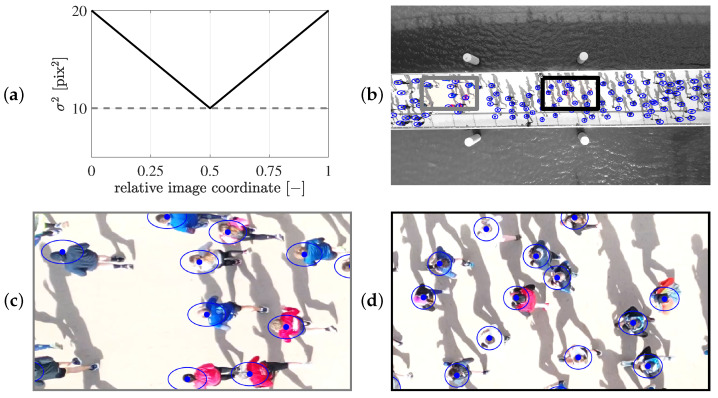
Determination of the random measurement covariance matrix in the image plane using an object detector. (**a**) Variance of the measurement error as a function of the relative horizontal (solid black line) and relative vertical image coordinate (dashed grey line). (**b**) Frame of a running event, highlighting a box at the edge (grey, corresponding to (**c**)) and at the centre of the image (black, corresponding to (**d**)). (**b**–**d**) 95% confidence regions are displayed as blue circles, with the detection indicated by a blue marker.

## Data Availability

Data will be made available from the corresponding author upon reasonable request as well as the Matlab R2023a processing scripts.

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
