# Peer review of "Sensors2025, 25(24), 7577;https://doi.org/10.3390/s25247577"

_sensors, 2025, doi:10.3390/s25247577_

Round 1

Reviewer 1 Report

Comments and Suggestions for Authors

This paper presents a dual-pipeline methodology for vision-based trajectory reconstruction in human activities, addressing two distinct scenarios: (i) controlled environments with small instrumented groups, and (ii) large-scale open-set crowd scenes. The first pipeline uses marker-based detection with colored hats and a blob-tracking strategy; the second combines Structure-from-Motion (SfM) scene reconstruction, camera calibration, open-set pedestrian detection (Grounding-DINO), and KLT-based tracking. The authors further integrate a Rauch–Tung–Striebel (RTS) smoother, homography projection, and a multi-stage error analysis. Experiments include two controlled studies and two major running events (Brussels 20 km and Antwerp 10 Miles), reporting position errors down to 0.10–0.15 m for the instrumented pipeline and 0.2–0.5 m for the large-scale pipeline. The work is relevant and practically useful; however, several major revisions are required to improve the manuscript’s clarity, reproducibility, and rigor.
1- The authors state they used MATLAB R2023a and Agisoft Metashape, but they do not provide complete implementation details. All preprocessing steps, detection routines, tracking modules, SfM/BA parameters, Kalman/RTS parameters, and covariance-estimation procedures must be made available or fully documented. This should include exact version numbers, toolboxes, data formats, and execution scripts.
2- The manuscript omits important details on image resizing and normalization, morphological operations used for blob extraction, KLT parameters (number of features, pyramid levels, window sizes), Grounding-DINO thresholds (text and box thresholds of 0.1 and 0.3 are mentioned but not justified), IoU/NMS thresholds, smoothing parameters, and covariance initialization. The authors should list all hyperparameters, thresholds, and runtime settings; a complete table of hyperparameters would be useful.
3- The authors must indicate the number of independent runs (random seeds) used for stochastic components (trained detectors, KLT, EM for covariance, bundle adjustment) and report metrics with uncertainties (± standard deviation or 95% CI) for the main measures (reprojection error, world-position error, missing detections). While orders of magnitude may be shown in the manuscript (e.g., 0.10–0.15 m for the instrumented pipeline, 0.20–0.50 m for the large-scale pipeline), these figures must be accompanied by uncertainty estimates and a clear statistical protocol.
4- The manuscript uses a fixed offset (mean BELHES = 1.7 m) for the homography plane and documents its impact on measurements (height distribution in the sample, effects on estimated velocities, systematic bias). The authors should add an ablative test using a realistic height distribution (not just ±0.1/0.2 m) and, if possible, propose an adaptive correction (for example, estimate apparent height via blob aspect-ratio or incorporate height uncertainty into the filter covariance). In addition to the current height-variation simulations, these analyses should be extended and the supporting scripts better explained.
4- The authors must compare the proposed pipelines against one or two baselines (for example, OpenPose or DeepLabCut for small groups, and a conventional closed-set detector for continuous streams) on a subset of the instrumented events. They should report detection accuracy/recall/F1, RMSE (in meters) for reconstructed trajectories, and the percentage of lost or merged tracks. Although these tools are mentioned in the text, empirical comparisons are required.

Reviewer 2 Report

Comments and Suggestions for Authors

This paper describes a vision-based trajectory reconstruction capable of modelling the movement of small and large groups of people. Important applications in civil engineering are described. It is an interesting work. The methodology appears sound, and the paper is well-written.

Moreover, the authors have collected significant datasets including thousands of participants.  If this could be made publicly available, rather than by request, I believe it could be very useful for the research community.

Section 1.2, you describe wearable and optical sensors. Are depth sensors also an option?

Section 2.1, in both recording experiments, are participants all moving in just one direction? Were any experiments performed with normal walking?

2.2.3. “When participant instrumentation was feasible, each person was provided with a coloured hat,”. Was this for all experiments? From earlier descriptions, I thought that this was just for the running track. Please clarify. What is feasible and not feasible? Please be specific.

Earlier, you described the limitations of other collection methods, such as OpenPose, which you claim has issues in uncontrolled environments. Does the requirement of coloured hats limit the real-world applications?

In the workflow in Figure 4, am I correct in thinking that either a moving or a stationary camera is used, not both? If so, I think this should be separated into two workflows (though they can remain in the same figure).

Figures 5 and 6 show high-contrast backgrounds (red or white surface) with colourfully clothed participants. Considering that the system uses colour-based segmentation, how would this work in more realistic environments?

Although Human Pose Estimation is briefly mentioned/discounted in Section 1.2, I wonder if there is any potential for future work to extract additional keypoints, such as bodily joints, to provide additional information regarding walking patterns and synchronisation.

Topham, L.K., Khan, W., Al-Jumeily, D. and Hussain, A., 2022. Human body pose estimation for gait identification: A comprehensive survey of datasets and models. ACM computing surveys55(6), pp.1-42.

Similarly, we know that gait may be used as a biometric; therefore, is the data extracted during this process identifiable? Should de-identification approaches be applied?

Directions for future work should be suggested.

Round 2

Reviewer 1 Report

Comments and Suggestions for Authors

The authors have applied all the proposed recommendations.